# Graph Contrastive Learning Meets Graph Meta Learning: A Unified Method for Few-shot Node Tasks

## ABSTRACT

Graph Neural Networks (GNNs) have become popular tools for Graph Representation Learning (GRL). One fundamental problem is few-shot node classification. Most existing methods follow the meta learning paradigm, showing the ability of fast generalization to few-shot tasks. However, recent works indicate that graph contrastive learning combined with fine-tuning can significantly outperform meta learning methods. Despite the empirical success, there is limited understanding of the reasons behind it. In our study, we first identify two crucial advantages of contrastive learning over meta learning, including (1) the comprehensive utilization of graph nodes and (2) the power of graph augmentations. To integrate the strength of both contrastive learning and meta learning on the few-shot node classification tasks, we introduce a new paradigm—**Co**ntrastive Few-Shot Node C**la**ssification (**COLA**). Specifically, COLA identifies semantically similar nodes only from augmented graphs, enabling the construction of meta-tasks without label information. Therefore, COLA can incorporate all nodes to construct meta-tasks, reducing the risk of overfitting. Through extensive experiments, we validate the necessity of each component in our design and demonstrate that COLA achieves new state-of-the-art on all tasks.

## CCS CONCEPTS

• **Computing methodologies → Machine learning**.

## KEYWORDS

Few-shot learning, Node classification, Unsupervised learning

**ACM Reference Format:**

Anonymous Author(s). 2023. Graph Contrastive Learning Meets Graph Meta Learning: A Unified Method for Few-shot Node Tasks. In *Proceedings of ACM Conference (Conference'17)*. ACM, New York, NY, USA, 15 pages. https://doi.org/10.1145/nnnnnnn.nnnnnnn

## 1 INTRODUCTION

Graph Neural Networks (GNNs) [10, 17] have emerged as the predominant encoders for Graph Representation Learning (GRL) in modern research, with node classification standing out as an essential domain of exploration. While a significant portion of the study has centered on employing GNNs in supervised or semi-supervised

contexts [33, 36], these approaches often require abundant annotated data. Nevertheless, acquiring high-quality labels is challenging in many scenarios, leading to a growing interest in exploring few-shot transductive node classification (FSNC), where only limited labeled samples are provided for novel classes. Real-world applications of FSNC span areas like classifying papers by topic from emerging research fields in a citation network [37] and categorizing newly introduced products within a co-purchasing network [26].

The majority of current study on FSNC [5, 13, 18, 20, 25, 35, 40] is rooted in the meta learning paradigm [6, 27]. Essentially, meta learning creates a series of meta-tasks during training to emulate real-world few-shot scenarios. To tackle a few-shot problem with $N$ classes and $k$ samples per class, meta learning iteratively trains over numerous $N$-way $k$-shot meta tasks derived from the training classes. Each meta-task consists of a support set and a query set, sampled from nodes belonging to a fixed number ($N$) of classes. The objective is to develop an algorithm that can perform well on the query set by training on only a few support samples. By constructing and resolving meta-tasks iteratively, models can learn the latent task distribution and adapt to tasks with novel classes.

Another emerging trend to effectively handle few-shot tasks is contrastive learning (CL). CL leverages positive and negative sample pairs to learn embeddings such that similar samples are brought closer in the embedding space while dissimilar ones are pushed apart. Several studies [32] have underscored the importance of transferable and discriminative representations for few-shot tasks. Observing the success of CL in other domains like computer vision [2, 3, 9], a recent exploration [30] on FSNC used pre-trained node embeddings learned from existing Graph Contrastive Learning (GCL) methods [15, 21] to train a linear classifier for few-shot tasks. This strategy has achieved notable success even without label information, surpassing previous state-of-the-art (SOTA) performance established by conventional meta learning methods.

To understand the success behind CL, we analyze and validate two critical factors contributing to contrastive learning's exceptional performance. The first factor is the use of data augmentation. By maximizing the similarity between a data point and its augmented version, CL ensures the model learns discriminative embeddings with minimal redundant information from the graph, which is essential for few-shot tasks. Secondly, CL's self-supervised learning (SSL) nature becomes especially powerful within the framework of FSNC and its transductive setting because label information is ignored in SSL. This enables CL to incorporate information of **all graph nodes** beyond the labeled ones, largely increasing the training sample size. In contrast, meta learning inherently relies on label information. Consequently, it can only include **nodes from the training classes** in meta-tasks, losing a significant portion of graph information. Furthermore, when the number of available training classes is limited, the meta-tasks may lack the necessary diversity to ensure robust generalization across few-shot scenarios.

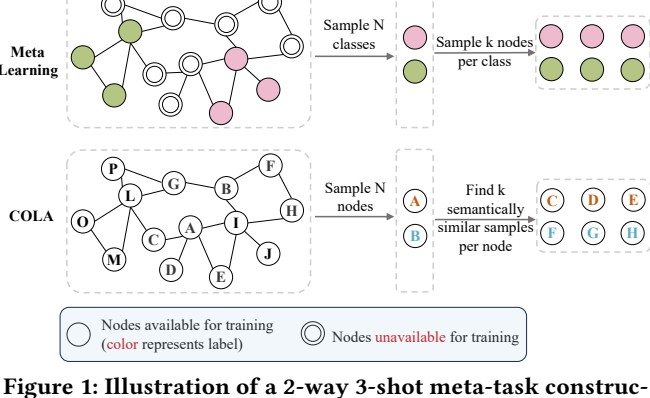

**Figure 1: Illustration of a 2-way 3-shot meta-task construction. COLA can leverage all nodes in the graph to construct meta-tasks, while previous meta learning methods can only use nodes from training classes.**

Hence, one natural question emerges: *Can we leverage the advantages of contrastive learning to enhance the current meta learning framework?* To address this question, we propose a new paradigm for few-shot node classification termed **Co**ntrastive Few-Shot Node **Cla**ssification (**COLA**). Unlike original meta-tasks, which require nodes within the same class to construct support sets, COLA constructs meta-tasks **without labels** (as illustrated in Figure 1). Hence, COLA can use all node information like CL can while benefiting from the few-shot-oriented meta learning framework.

Creating support and query sets is the core of $N$-way $k$-shot meta-tasks construction. Inspired by self-supervised contrastive learning, COLA randomly samples $N$ query nodes, with each representing one way in meta learning. The main challenge is constructing the support set including $k$ samples semantically similar to each query node without label information. To achieve this, we use GNNs to get node embeddings from three augmented graphs. Given a query node, we first obtain its embedding from the first graph. If we find a set of embeddings from the second graph that matches the query embedding closely, then this set should maintain high similarity with the query node's embedding from the third graph. We then treat this embedding set as the support set and maximize the similarity between this set and the query embedding from the third graph.

Our framework has several advantages: (1) We propose a novel method that utilizes the invariant information among three augmented graphs to construct semantically correct meta-tasks without label information; (2) Unlike conventional meta-tasks constructed based on training labels, COLA meta-tasks are based on semantic similarity, preventing overfitting to training classes. (3) COLA meta-tasks use all nodes in training, incorporating much more graph information than traditional meta learning methods that only use labeled nodes. We conduct extensive experiments on seven real-world datasets and examine the necessity of each component in our framework. Our results demonstrate that COLA outperforms all previous methods, achieving new state-of-the-art performance on few-shot node classification. COLA's outstanding performance demonstrates that meta-learning remains a powerful solution for few-shot tasks when all graph nodes are used.

## 2 RELATED WORK

**Graph Few-shot Learning.** While GNNs for node classification are generally semi-supervised [17], considerable efforts were spent on removing the labeling dependency [10, 29, 33]. However, they cannot handle unseen classes during the test phase. This inspired research on the few-shot node classification problem. The majority of research employs a meta learning paradigm. Meta-GNN [40] adapts the optimization-based meta learning method MAML [6] to graph data. GFL [38] enables few-shot classification on unseen graphs with seen node classes. GPN [5] uses ProtoNet [27], a metric-based meta learning method, and refines prototypes with the weights learned by a GCN [17]. G-Meta [13] leverages subgraph information and achieves good performance on both transductive and inductive FSNC. RALE [20] assigns relative and absolute locations to each node within meta-tasks. TENT [35] applies node-level, class-level, and task-level adaptations in each task to mitigate task variance impact. Recently, TLP [30], inspired by graph contrastive learning, trains a few-shot classifier using pre-trained node embeddings, thereby significantly enhancing the performance over existing meta learning approaches. Its success prompts us to delve further into the potential of contrastive learning.

**Graph Contrastive Learning.** Contrastive Learning methods [2, 3, 9] have been adapted to the graph domain. DGI [34] learns node representations by maximizing mutual information (MI) between local and global graph features. GRACE [41] maximizes node-level agreement between two corrupted views. MVGRL [11] maximizes the MI between node representations of one view and graph representations of another view. GraphCL [39] applies various data augmentation techniques to the graph and then employs a contrastive loss function to move the representations of augmented views of the same graph closer. MERIT [15] leverages bootstrapping within a Siamese network and multi-scale graph contrastive learning to enhance node representation learning. SUGRL [21] employs node embeddings from MLP as anchors and takes advantage of structural and neighbor information to obtain two kinds of positive samples. Different from previous methods, SUGRL takes the combination of triplet loss instead of InfoNCE loss [23]. BGRL [31] extends the non-contrastive setting [9] that does not need negative samples to graph problems. TLP [30] trains a linear classifier on top of embedding learned from various graph contrastive learning methods, where SUGRL consistently delivers superior performance on few-shot tasks.

**Few-shot Learning with Contrastive Learning.** Recent works in computer vision show that meta learning and contrastive learning can benefit from each other. Some recent few-shot auxiliary learning works [4, 8, 28] view few-shot learning as the main task and combine the few-shot loss with self-supervised auxiliary tasks. Liu et al. [19] employs supervised contrastive learning on meta-tasks, where support images and query images are processed with designed data augmentations to construct hard samples. CPLAE [7] represents support and query samples using concatenated embeddings of both the original and augmented versions. It then regards prototypes of support samples as the anchor samples in contrastive learning. However, these methods are suboptimal on graph tasks due to their reliance on label information and domain-specific data augmentations. PsCo [14] employs a Moco[3] inspired momentum

network with a queue system, aiming to improve the diversity of meta-tasks in the unsupervised meta learning setting. However, the queue-like setup is redundant in graph-based node classification where each batch provides embeddings for all nodes in the graph. MetaContrastive [22] proposes a meta learning framework to enhance contrastive learning by transforming contrastive learning setup to meta-tasks. **Notably, in the field of graph learning, there is no work that enhances meta learning with the advantages of contrastive learning, and it is challenging to tailor these previous methods from the image domain for the graph.** We also provide experimental results in Table 3 by adapting some works in the image domain to FSNC to validate the assertion.

## 3 NOTATIONS AND PRELIMINARIES

We first introduce some preliminary concepts and notations. In this work, we consider an undirected attributed graph $\mathcal{G} = (\mathcal{V}, \mathcal{E}, \mathbf{A}, X)$, where $\mathcal{V} = \{v_1, \cdots, v_{|\mathcal{V}|}\}$ is the set of nodes, $\mathcal{E} = \{e_1, \cdots, e_{|\mathcal{E}|}\}$ is the set of edges. The adjacency matrix $\mathbf{A} \in \{0, 1\}^{|\mathcal{V}| \times |\mathcal{V}|}$ describes the graph structure, with $\mathbf{A}_{ij} = 1$ indicating an edge between nodes $v_i$ and $v_j$ and $\mathbf{A}_{ij} = 0$ otherwise. The feature matrix $X \in \mathbb{R}^{|\mathcal{V}| \times d}$ contains the node features, where $\mathbf{x}_i \in \mathbb{R}^d$ represents the feature of node $v_i$ and $d$ is the feature dimension. Our work focuses on the node classification problem, where each node $i$ has a label $y_i \in C$ and $C$ is the set of labels with $|C|$ different classes.

**Few-shot Node Classification.** In node classification, nodes are usually divided into train, validation, and test sets, denoted as $X_{train}$, $X_{val}$, and $X_{test}$, respectively. However, unlike supervised node classification where the node labels of train/validation/test sets are sampled from the same label set $C$, the label of nodes in few-shot learning are sampled from non-overlapped label sets for train/validation/test set, denoted as $C_{train}$, $C_{val}$. and $C_{test}$. Further, it holds that $C_{train} \cap C_{test} = \emptyset$. Few-shot Learning typically deals with $N$-way $k$-shot tasks, where the objective is to classify nodes into one of $N$ distinct classes using only $k$ labeled samples per class.

**Meta Learning.** Meta learning [6, 27] tries to solve the few-shot problems by designing a novel training strategy. The overall process of meta learning can be divided into meta-train and meta-test phases. During meta-train phase, the model is trained to simulate the few-shot learning environment. It enables the model to quickly adapt to new few-shot tasks with limited labeled data during the meta-test phase. Specifically, at each training episode, meta learning constructs $N$-way $k$-shot tasks using samples from the training set $X_{train}$. To form an $N$-way $k$-shot task, meta learning first randomly select a class set $C_{meta}$ with $N$ classes from $C_{train}$ and then generate a **support set** $\mathcal{S} = \{(\mathbf{x}_i, y_j) | y_j \in C_{meta}, i = 1, \cdots, N \times k\}$ and a **query set** $Q = \{(\mathbf{x}_i, y_j) | y_j \in C_{meta}, i = 1, \cdots, N \times q\}(\mathcal{S} \cap Q = \emptyset)$ by sampling $k$ support and $q$ query samples from each class in $C_{meta}$, respectively. The objective is to train on the support set so that it can perform well on the query set. In meta-test phase, the $N$-way $k$-shot tasks are constructed with samples in $X_{test}$ in a similar way.

## 4 CONTRASTIVE FEW-SHOT NODE CLASSIFICATION (COLA)

In this section, we first identify two critical components that contribute hugely to the success of contrastive learning on FSNC but

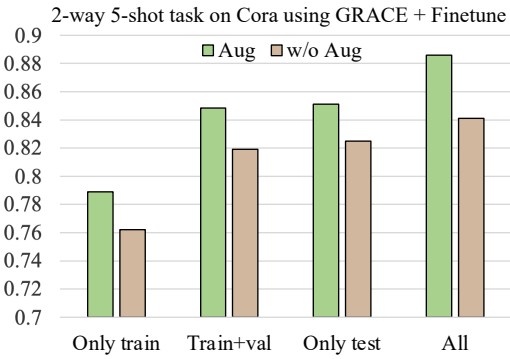

**Figure 2: 2-way 5-shot task on Cora using GRACE+finetune. Accuracy of four situations w/ and w/o augmentations.**

are not present in meta learning. Then, we introduce a new paradigm COLA, which leverages the strengths of both contrastive learning and meta learning. The key idea is to construct meta-tasks without labels. We use the invariant information among three augmented graphs to construct semantically correct meta-tasks. We then take the supervised contrastive loss to learn the meta-tasks.

### 4.1 Analysis on Success of Contrastive Learning in Few-Shot Node Classification

Although most current works on transductive FSNC follow the meta learning framework (details discussed in Section 2), a recent study TLP [30] highlights the effectiveness of graph contrastive learning combined with fine-tuning. The authors conducted experiments using various existing graph contrastive learning methods and fine-tuned a linear classifier on top of the learned representation, which resulted in significant performance improvements on few-shot node classification tasks compared to SOTA supervised meta learning methods.

To understand the strong performance of contrastive learning, we analyze the difference between contrastive learning and meta learning. Both techniques strive to bring the embeddings of semantically similar nodes closer and separate embeddings of semantically dissimilar ones. However, the definition of semantical similarity is different in the two methods. Meta learning regards **all node embeddings from the same class as similar**, while those from different classes as dissimilar. In contrast, self-supervised contrastive learning only considers the embeddings of **the same node in different augmented graphs as similar**.

Such a definition of similarity provides contrastive learning with a distinct advantage in transductive FSNC problems. It allows the model to utilize all node embeddings in a given graph explicitly. Conversely, meta learning relies on labels from training classes $C_{train}$, thus, only nodes from these classes are involved in the meta-train phase, increasing the likelihood of overfitting to the training classes and limiting the model's ability to transfer knowledge to test classes. Especially when the number of training classes is insufficient, the diversity of meta-tasks is not guaranteed, thus hurting the generalization ability of meta learning. Further, leveraging the graph augmentation technique is another difference between contrastive learning and meta learning, which is already known to be

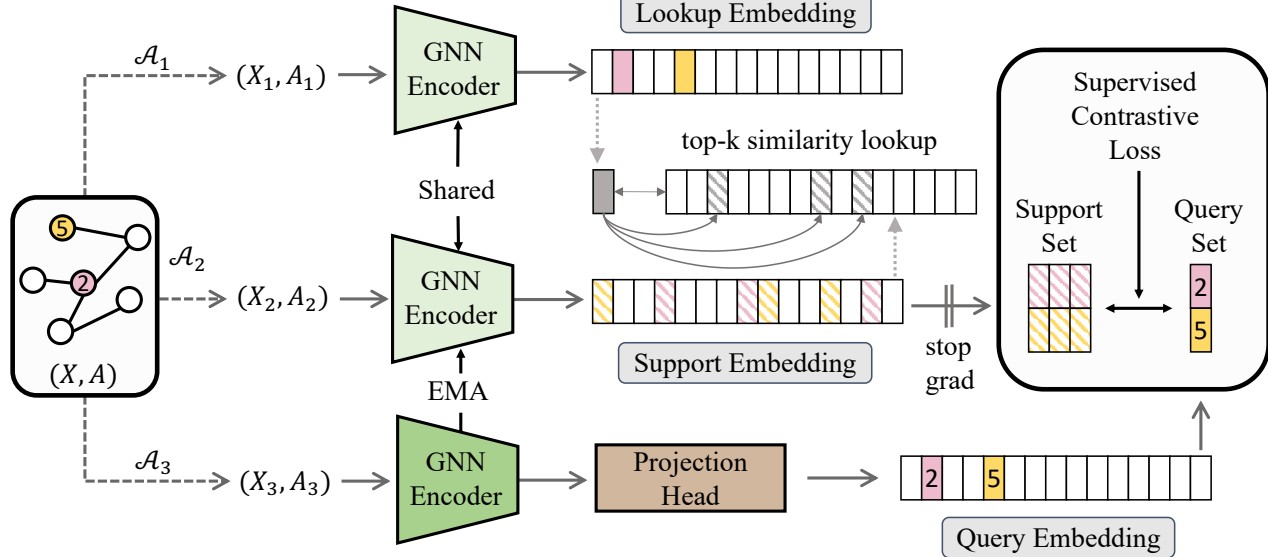

**Figure 3: An overview of the COLA Framework. The construction of a 2-way 3-shot meta-task is illustrated. Two nodes 2 and 5 are sampled as the query set. The query node's embedding in Lookup Embedding matches with all node embeddings in Support Embedding. Top-$k$ similar embeddings are selected for the support set. Supervised contrastive loss is calculated for each task.**

effective in learning discriminative representation [32]. We conjecture the above two differences contribute most to the success of contrastive learning in FSNC.

We then conduct extensive ablation studies to validate our hypothesis. We present one experimental result in Figure 2 and include other results in Appendix C. The experiment is conducted on a 2-way 5-shot task from Cora [37] dataset, and the node embeddings pre-trained from a GCL model named GRACE [41] are used to train a classifier for few-shot tasks. We control the nodes used for pre-training to be sampled from $C_{train}$, $C_{train} \cup C_{val}$, $C_{test}$, and the whole graph. $C_{train}$, $C_{val}$ and $C_{test}$ contain 3, 2, 2 non-overlapped classes, respectively. We then assess the model on few-shot tasks sampled from $C_{test}$.

The results reveal several insights: although the number of nodes belonging to $C_{train} \cup C_{val}$ far exceeds the number of nodes in $C_{test}$, only using samples from $C_{test}$ to pretrain achieves better result than the other two settings. Note that the label information is not included in the pretraining process. This experiment validates that explicitly leveraging test class samples during training can avoid overfitting. Besides, using all nodes can maximize the utilization of graph information. Another observation is that eliminating augmentation leads to a performance decrease. Thus, the discriminative representation acquired by contrastive learning through data augmentation techniques is also crucial for few-shot tasks.

From experimental results, we can see the explicit use of all nodes and data augmentation are crucial to contrastive learning performance. These insights inspire us to propose a more robust meta learning framework that can effectively leverage these advantages of contrastive learning while also benefiting from the generalization capabilities of meta learning.

## 4.2 Meta-task Construction without Labels

In this section, we introduce our framework COLA, and the overall framework is illustrated in Figure 3 and Algorithm 1. COLA aims to construct meta-tasks without labels, such that all nodes can be explicitly used during training.

Inspired by contrastive learning, we first sample $N$ nodes and regard them as $N$ distinct classes to form the query set $Q$. Denote the query set $Q = \{v_1, \cdots, v_N\}$, where $v_i$ is the query node of the $i$-th way. To construct an $N$-way $k$-shot meta-task, the support set $S$ should include $k$ samples that have similar semantics to the query sample from each of the $N$ ways. Then, **how to find semantically similar samples is the main challenge**.

One naive idea is to use a GNN to get node embeddings and select nodes with top-$k$ similar embeddings to the query node's embedding as its support set. However, the support nodes selected in this way will only enlarge the similarity between nodes which the GNN initially regards similar—if the GNN happens to give two semantically distinct nodes similar embeddings at first, such a way will update the model so that their embeddings are more and more similar, ultimately collapsing the model. Inspired by CL, one remedy is to take the query node's embedding from one augmented view and match it against the embeddings of all nodes from another view to select the top-$k$ similar support nodes. This solution enhances the robustness of the selection as nodes with similar embeddings across different augmented graphs are more likely those "truly" semantically similar ones. However, it still does not solve the model collapse risk, meaning an initially bad GNN gets worse and worse.

We thus introduce a **third** augmented view and a **momentum GNN** to ensure a more robust and comprehensive meta-task construction. The core idea is as follows. We first generate query node

embeddings from one augmented view. To select semantically similar support nodes, we do not compare to these query node embeddings directly. Instead, we additionally generate a lookup embedding for each query node in a second augmented graph, and match these lookup embeddings with support embeddings generated from a third augmented graph to select the support nodes (shown in Figure 3). The three different views greatly enhance the robustness of the meta-task construction. Furthermore, the lookup embeddings and support embeddings are obtained from a momentum GNN instead of the GNN to train (the one used to get the query embeddings), so that model collapse is prevented. Finally, the GNN is trained to maximize the similarity between the query embedding and the support embeddings of the selected support nodes. We empirically verified all our designs via a thorough ablation study (see Section 5.3). Below we detail the meta-task construction process.

For a graph $\mathcal{G}$, let $\mathcal{A}(\mathcal{G})$ denote the distribution of graph data augmentation of $\mathcal{G}$. These augmentations [39] typically involve one or more operations, such as node dropping, edge perturbation, and attribute masking. For the given graph represented as $(X, \mathbf{A})$, we apply three different data augmentations $\mathbf{A}_1, \mathbf{A}_2, \mathbf{A}_3 \sim \mathcal{A}$ and generate the corresponding augmented graphs $(X_1, \mathbf{A}_1), (X_2, \mathbf{A}_2), (X_3, \mathbf{A}_3)$.

We then use GNNs to generate Lookup, Support, and Query Embeddings from the augmented graphs. Formally,

$$L := f_{\text{ema}}(X_1, \mathbf{A}_1), \ S := f_{\text{ema}}(X_2, \mathbf{A}_2), \ Q := g(f(X_3, \mathbf{A}_3)), \quad (1)$$

where Lookup Embedding $L$ and Support Embedding $S$ are generated by a momentum encoder $f_{\text{ema}}$, and the Query Embedding $Q$ is generated by a trainable graph encoder $f$ with a projection head $g$. Weights of $f_{\text{ema}}$ are the moving average from $f$. Details about the momentum encoder will be discussed later.

Then we present the process of constructing meta-tasks. We first get query nodes' embeddings from Lookup Embedding $L$ and denote them as $\{L_{v_1}, \cdots, L_{v_N}\}$. For each $i \in [1, \cdots, N]$, we then measure the similarity between $L_{v_i}$ and all node embeddings $\{S_1, \cdots, S_{|\mathcal{V}|}\}$ in Support Embedding $S$. The $k$ embeddings in $S$ with the highest similarity scores will be selected as the support set, leading to $Nk$ samples in the support set. We denote them as $\{S_{v_i^1}, \cdots, S_{v_i^k}\}_{i=1}^N$, where $S_{v_i^j}$ is the $j$-th support sample of the $i$-th query node. Finally, we get query nodes' embedding from Query Embedding $Q$ and denote them as $\{Q_{v_1}, \cdots, Q_{v_N}\}$ and use them as the query set to construct a meta-task together with the support set. The task $\mathcal{T}$ can be represented as $\mathcal{T} = \{Q_{v_i}, \{S_{v_i^j}\}_{j=1}^k\}_{i=1}^N$.

Our method uses the fact that the most essential graph information should be invariant across different augmented views. **Given a query node $v_i$, if we find $k$ embeddings (in $S$) that are very similar to $v_i$'s embedding from one augmented view ($L$), then these $k$ embeddings should also be closely similar to $v_i$'s embedding from a different augmented view $Q$.** Leveraging such invariance leads to more robust meta-task construction. To further verify the importance of each of the three embeddings, we have carried out comprehensive ablation studies presented in Section 5.3.1.

The momentum encoder is another important component of our meta-task construction. Formally, denote the parameters of $f_{\text{ema}}$ by $\theta_{\text{ema}}$ and parameters of $f$ by $\theta$, $\theta_{\text{ema}}$ is updated by exponential moving average (EMA): $\theta_{\text{ema}} = m\theta_{\text{ema}} + (1 - m)\theta$, where $m$ is the momentum coefficient to control what degree it preserves the

---

**Algorithm 1** COLA Meta-task Construction

---

**Require:** $f$: GNN encoder, $g$ : projection head, $f_{\text{ema}}$: momentum GNN encoder, $X$: feature matrix, $A$: adjacency matrix, $|V|$: number of nodes, $N$: number of classes, $k$: number of samples in support set, $d$: embedding dimension, $T$: number of meta-tasks.

1: Generate three views: $(X_1, A_1)$, $(X_2, A_2)$, $(X_3, A_3)$;
2: Compute Query $(Q)$, Support $(S)$, Lookup $(L)$ Embeddings by Eq(1);
3: Randomly sample $N$ nodes to construct query set;
4: **for** each node $v_i$ in query set **do**
5:      Get query node's embedding $L_{v_i}$ from $L$;
6:      Compute cosine similarity between $L_{v_i}$ and all node embeddings in $S$;
7:      Regard the $k$ embeddings with the highest similarity score as support set;
8:      Get query node's embedding $Q_{v_q}$ from Q;
9: **end for**
10: Compute contrastive loss $L_{COLA}$ using Eq( 2).

---

history. By employing a momentum encoder instead of the same trainable GNN encoder, the support set candidate pool ($S$) remains consistent across episodes and is less susceptible to noise or non-informative information from the rapidly changing encoder. Lookup Embedding and Support Embedding share the same momentum encoder, allowing for more accurate and consistent matching. The importance of the momentum encoder is validated in Section 5.3.2.

### 4.3 Training Procedure

**Meta-Train Phase.** To train the model, we want to maximize the similarity between the query embedding and corresponding support embeddings, thus we design the loss function inspired by the supervised contrastive loss [16]. In our setting, for each way $i$, the query embedding is treated as the anchor sample. The support embeddings $\{S_{v_i^1}, \cdots, S_{v_i^k}\}$ are considered as positive samples, while support embeddings $\{S_{v_{i'}^1}, \cdots, S_{v_{i'}^k}\}_{i' \neq i}$ from other ways are viewed as negative samples. Formally, the pseudo-supervised contrastive loss for each meta-task can be expressed as follows:

$$L_{COLA}(\{Q_{v_i}, \{S_{v_i^j}\}_{j=1}^k\}_{i=1}^N) = -\sum_{i=1}^N \frac{1}{k} \sum_{j=1}^k \log \frac{\exp(Q_{v_i} \cdot S_{v_i^j}/\tau)}{\sum_{\mathbf{v} \in S_t} \exp(Q_{v_i} \cdot \mathbf{v}/\tau)},$$
$$(2)$$

where $Q_{v_i}$ is the query sample of the $i$-th way, and $S_{v_i^j}$ is the $j$-th support sample of $Q_{v_i}$. $S_t$ denotes all the support embeddings in the current meta-task and $\tau$ is the temperature parameter. Finally, the loss function of each meta-train episode is the average loss of multiple meta-tasks.

**Meta-Test Phase.** During the meta-test phase, we discard the momentum encoder and retain the GNN encoder. Then a linear classifier is trained on top of the learned node embeddings from the GNN encoder. To elaborate, we initially select $N$ classes from $C_{test}$ and sample $k$ labeled nodes from each class. The embeddings of these samples then undergo supervised training to fit a linear classifier. In the final step, we evaluate the performance using $q$ nodes from each of the $N$ classes.

**Table 1: Results on Cora, CiteSeer, CoraFull, and Roman-Empire datasets. (Top rows) Meta Learning. (Middle rows) Graph Contrastive Learning with fine-tuning. (Bottom row) COLA (our method). All scores are averaged over 20 runs. Evaluation metrics were scaled to 100 for readability purposes. In bold are methods with the best results for each task. In blue are methods with the best results in each group.**

| Dataset | Cora | | CiteSeer | | CoraFull | | Roman-Empire | |
|---|---|---|---|---|---|---|---|---|
| Task | 2-way 1-shot | 2-way 5-shot | 2-way 1-shot | 2-way 5-shot | 5-way 1-shot | 5-way 5-shot | 2-way 1-shot | 2-way 5-shot |
| Meta learning | | | | | | | | |
| MAML [6] | $52.59 \pm 2.28$ | $56.45 \pm 2.41$ | $51.77 \pm 2.28$ | $54.21 \pm 2.30$ | $22.47 \pm 1.21$ | $26.58 \pm 1.32$ | $53.65 \pm 2.54$ | $55.71 \pm 2.36$ |
| ProtoNet [27] | $51.69 \pm 2.17$ | $55.00 \pm 2.39$ | $51.43 \pm 2.12$ | $53.23 \pm 2.28$ | $34.17 \pm 1.74$ | $46.86 \pm 1.74$ | $53.73 \pm 2.16$ | $56.49 \pm 2.03$ |
| Meta-GNN [40] | $57.87 \pm 2.52$ | $57.35 \pm 2.30$ | $55.12 \pm 2.62$ | $60.59 \pm 3.26$ | $55.36 \pm 2.49$ | $71.42 \pm 2.02$ | $54.59 \pm 2.70$ | $59.31 \pm 2.43$ |
| GPN [5] | $56.09 \pm 2.08$ | $63.83 \pm 2.86$ | $59.33 \pm 2.23$ | $65.60 \pm 2.47$ | $56.48 \pm 2.72$ | $71.23 \pm 2.11$ | $58.42 \pm 2.31$ | $63.96 \pm 2.27$ |
| G-Meta [13] | $66.15 \pm 3.00$ | $82.85 \pm 1.19$ | $54.33 \pm 2.02$ | $61.47 \pm 2.37$ | $58.47 \pm 2.37$ | $72.03 \pm 1.88$ | $61.45 \pm 2.35$ | $62.99 \pm 2.58$ |
| TENT [35] | $54.33 \pm 2.10$ | $58.97 \pm 2.40$ | $60.06 \pm 3.01$ | $66.31 \pm 2.45$ | $49.83 \pm 2.02$ | $64.23 \pm 1.75$ | $60.32 \pm 2.14$ | $67.14 \pm 1.95$ |
| Graph Contrastive Learning + Finetune | | | | | | | | |
| BGRL [31] | $59.16 \pm 2.48$ | $81.31 \pm 1.89$ | $54.33 \pm 2.14$ | $66.74 \pm 2.13$ | $40.82 \pm 1.95$ | $69.98 \pm 1.67$ | $56.58 \pm 2.31$ | $66.39 \pm 1.74$ |
| MVGRL [11] | $74.96 \pm 2.94$ | $91.32 \pm 1.47$ | $63.39 \pm 2.69$ | $79.73 \pm 1.92$ | $66.40 \pm 2.31$ | $83.99 \pm 1.51$ | $64.32 \pm 2.77$ | $71.74 \pm 1.69$ |
| MERIT [15] | $70.63 \pm 3.11$ | $91.00 \pm 1.22$ | $65.64 \pm 2.94$ | $78.54 \pm 2.43$ | $65.17 \pm 1.96$ | $84.74 \pm 1.44$ | $64.83 \pm 2.81$ | $74.16 \pm 1.74$ |
| GraphCL [39] | $74.32 \pm 3.26$ | $90.43 \pm 1.21$ | $71.39 \pm 3.17$ | $79.60 \pm 1.89$ | $66.76 \pm 2.75$ | $84.55 \pm 1.48$ | $65.13 \pm 2.98$ | $73.01 \pm 2.24$ |
| GRACE [41] | $71.50 \pm 1.42$ | $88.49 \pm 1.44$ | $67.43 \pm 2.51$ | $82.09 \pm 1.64$ | $62.05 \pm 2.22$ | $81.54 \pm 1.52$ | $62.78 \pm 1.76$ | $74.50 \pm 1.73$ |
| SUGRL [21] | $81.52 \pm 2.09$ | $92.49 \pm 1.02$ | $72.43 \pm 2.42$ | $86.58 \pm 1.19$ | $73.95 \pm 2.13$ | $83.07 \pm 1.21$ | $64.69 \pm 2.01$ | $73.05 \pm 1.69$ |
| COLA (ours) | $\mathbf{84.58 \pm 1.96}$ | $\mathbf{94.03 \pm 1.48}$ | $\mathbf{76.54 \pm 2.02}$ | $\mathbf{86.87 \pm 1.49}$ | $\mathbf{74.36 \pm 2.37}$ | $\mathbf{86.59 \pm 2.26}$ | $\mathbf{68.59 \pm 1.83}$ | $\mathbf{78.96 \pm 1.87}$ |

## 5 EXPERIMENT

In this section, we demonstrate COLA outperforms all the baselines in each task and provide the ablation study to validate the significance of each model component.

### 5.1 Datasets, Setup, and Baselines

*5.1.1 Datasets.* We conducted our experiments on seven benchmark datasets: Cora [37], CiteSeer [37], Amazon-Computer [26] (Computer), CoraFull [1], Coauthor-CS [26] (CS), ogbn-arxiv [12] and Roman-empire [24]. The first six are all commonly used homophilous graph datasets, and the last one is a heterophilous graph dataset. In each run for the same dataset, the classes were randomly divided into three subsets: $C_{train}$, $C_{val}$, and $C_{test}$. The setting of the split ratio follows previous works [30] and a detailed description of these datasets is provided in Appendix A.

*5.1.2 Implementation Details.* We utilized Graph Convolutional Networks [17] (GCNs) as the encoder for all homophilous graphs and used GraphSAGE [10] as the encoder for the heterphilous graph, and took a multi-layer perceptron (MLP) as the projection head. Our data augmentation combines edge and feature dropout. The number of training tasks for calculating the average loss function is set to 20. We report mean accuracy and the 95% confidence interval of 20 runs for both COLA and baseline models for a fair comparison. All models were tested on a single NVIDIA A100 80GB GPU. The detailed setting of hyperparameters and source code are provided in Appendix B.

*5.1.3 Baselines.* We mainly compared our model with two groups of baselines: meta learning and graph contrastive learning with finetuning (proposed by TLP [30]). For meta learning, we first evaluate two plain meta learning models without GNN [17] as backbone: MAML [6] and ProtoNet [27], then we evaluate several meta learning works for few-shot node classification: Meta-GNN [40], GPN [5], G-Meta [13], and TENT [35]. For TLP methods, we adhered to the settings and evaluated different graph contrastive learning methods for both contrastive-GCL and noncontrastive-GCL. They are MVGRL [11], GraphCL [39], GRACE [41], MERIT [15], SUGRL [21], and BGRL [31], respectively. Besides, to validate our assertion that the works from the image domain may show suboptimal performance when applied to FSNC, we apply two related works BFS [14] and PsCo [14] to the graph domain.

### 5.2 Main Results

Evaluations were made under 2-way 1-shot/5-shot settings on Cora, CiteSeer, Computer, and Roman-Empire datasets due to the limited number of available classes. CoraFull, Coauthor-CS, and ogbn-arxiv datasets were evaluated under 5-way 1-shot/5-shot settings. We present the main results on Cora, CiteSeer, CoraFull, and Roman-Empire datasets in Table 1 and include results on other datasets in Appendix C. The comparison between COLA and related works in the image domain is presented in Table 3.

**Our method COLA outperforms all the other baselines in every task.** Compared with meta learning methods, COLA achieves at least 11.18% and up to 20.56% absolute accuracy improvement. The results demonstrate that the utilization of all nodes and a

**Table 2: Component Analysis of Query (Q), Support (S), Lookup (L) Embeddings on Cora and CiteSeer datasets. The first three rows control different components in meta-task construction. The last row is COLA's setting. In bold are the best results, and underlines are the second best ones.**

| Q | S | L | Cora | | | CiteSeer | | |
|---|---|---|---|---|---|---|---|---|
| | | | 2-way 1-shot | 2-way 3-shot | 2-way 5-shot | 2-way 1-shot | 2-way 3-shot | 2-way 5-shot |
| ✓ | | | 61.90 ± 1.26 | 84.12 ± 2.24 | 88.24 ± 1.89 | 56.03 ± 1.73 | 71.46 ± 2.97 | 74.69 ± 2.22 |
| ✓ | ✓ | | 75.79 ± 2.75 | 75.20 ± 2.68 | 79.44 ± 2.01 | 59.48 ± 2.83 | 63.73 ± 2.48 | 69.10 ± 2.31 |
| ✓ | | ✓ | 76.24 ± 3.68 | 86.47 ± 1.45 | 85.78 ± 2.57 | 64.42 ± 2.34 | 69.33 ± 3.15 | 73.13 ± 2.27 |
| ✓ | ✓ | ✓ | **84.58 ± 1.96** | **92.29 ± 1.71** | **94.03 ± 1.48** | **76.54 ± 2.02** | **80.26 ± 2.72** | **86.87 ± 1.49** |

**Table 3: Comparison with related works in the image domain.**

| Dataset | Cora | | CoraFull | |
|---|---|---|---|---|
| Task | 2-way 1-shot | 2-way 5-shot | 5-way 1-shot | 5-way 5-shot |
| BFS [8] | 70.46 ± 2.22 | 90.57 ± 1.89 | 60.44 ± 2.75 | 82.31 ± 1.66 |
| PsCo [14] | 71.70 ± 1.85 | 90.31 ± 1.64 | 62.46 ± 2.40 | 81.37 ± 2.17 |
| COLA | **84.58 ± 1.96** | **94.03 ± 1.48** | **74.36 ± 2.37** | **86.59 ± 2.26** |

**Table 4: Relationship between the momentum parameter and accuracy on CiteSeer.**

| momentum | 0 | 0.5 | 0.8 | 0.9 | 1 |
|---|---|---|---|---|---|
| 2-way 1-shot | 70.46 | 74.47 | 75.13 | 76.54 | 54.85 |
| 2-way 5-shot | 78.05 | 81.09 | 83.34 | 86.87 | 65.43 |

discriminative data representation indeed benefit the learning on few-shot tasks. Thus even when constructing meta-tasks without label information, COLA can achieve excellent performance over traditional meta learning methods.

Graph contrastive learning methods benefit from the learned discriminative representations and show excellent ability to deal with downstream few-shot tasks. SUGRL achieves the best performance on most few-shot tasks. COLA outperforms SUGRL in each task with a maximum relative accuracy improvement of 8.09%. This demonstrates that the use of $N$-way $k$-shot task construction in COLA makes it more suitable for few-shot problems compared to contrastive learning methods.

Moreover, while studies from the image domain present results comparable to some graph contrastive learning benchmarks, COLA still significantly outperforms these methods. This validates our discussion on the challenges of directly adapting methodologies from the image domain to the graph context.

Note that, in the transductive setting adopted by all baselines, the full graph is given in advance and even meta learning methods still see and aggregate the test nodes. The difference is that they don't construct meta-tasks for these test nodes, while ours does, which leads to a more thorough utilization of the full graph information.

## 5.3 Model Design Component Analysis

*5.3.1 Query, Support, Lookup Embeddings.* First, we examine the primary design elements of COLA: Query ($Q$), Support ($S$), and Lookup Embeddings ($L$) and present the results in Table 2. To understand the distinct function of each one, we investigate three alternative scenarios. In the first scenario, we only use the Query Embedding. The query sample $Q_{v_i}$ is extracted from the Query Embedding and has to align with all nodes within the Query Embedding itself to identify the support set. The second scenario omits the Lookup Embedding. Here, query sample $Q_{v_i}$ is compared with all nodes from Support Embedding to find the top-$k$ similar ones in

$S$. The third scenario excludes Support Embedding, so we use the query embedding $L_{v_i}$ from Lookup Embedding to compare with all node embeddings in Query Embedding.

Compared with COLA, the first and second scenarios regard the query embedding as its own lookup tool. This reduces the amount of information in the meta-task, leading to suboptimal results. In the second scenario, the use of Support Embedding further deteriorates the performance since the inconsistency between the Query and Support Embeddings' encoders leads to a mismatch. The third scenario involves the Lookup Embedding, but both the query and support set are derived from Query Embedding, which means the model cannot take advantage of the extra information gained from two different augmented views. We also find that even some of these suboptimal setups can still outperform meta learning methods, underscoring the importance of using all available nodes.

Our COLA model significantly outperforms the three scenarios, illustrating the importance of each component in its design. In essence, we benefit from the invariant information among these three augmented graphs to construct meta-tasks, such that the support set selected by Lookup Embedding has very similar semantics to the query set. This ensures our model keeps constructing semantically correct meta-tasks.

*5.3.2 Momentum Encoder.* To generate Support and Lookup Embedding, COLA uses a momentum GNN encoder $f_{\text{ema}}$, whose weights are an exponential moving average of the weights of the trained GNN encoder. The momentum encoder is more stable than the trainable GNN encoder. We test different values of the momentum variable from 0 to 1 and present the results in Table 4. Value 0 means the encoder is updated to the trained GNN encoder at each step and value 1 means the encoder is never updated. The results show that using the shared weight encoder (momentum=0) will harm the model performance. A static encoder (momentum=1) always contains the exact same information and constrains the information support embeddings can bring. A larger momentum (around 0.9) can help the momentum encoder memorize historical information, contributing to a consistent and stable Support Embedding.

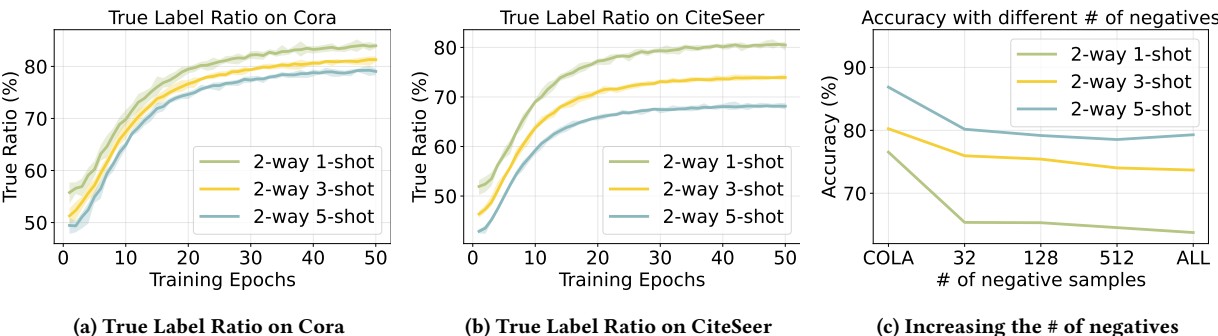

(a) True Label Ratio on Cora     (b) True Label Ratio on CiteSeer     (c) Increasing the # of negatives

Figure 4: (a) and (b): true label ratio that measures the ratio of the selected support samples actually having the same label as the query sample. (c): Performance drops with extra negative samples.

## 5.4 Deep Investigation of COLA

5.4.1 *True Label Ratio.* We propose use true label ratio to evaluate the quality of task construction. True label ratio is calculated by

$R_{true} = \frac{\sum_{i=1}^{N} \sum_{j=1}^{k} (y_{v_i^j} == y_{v_i})}{Nk}$, where $y_{v_i}$ is the label of query node $v_i$

and $y_{v_i^j}$ is the label of the $j$-th support node of $v_i$, $Nk$ is the total number of support samples. It aims to measure how many nodes in the support set have the same label with corresponding query node. To better visualize the trend, we only present the true label ratio within 50 epochs in Figure 4a and 4b. Note that $R_{true}$ still increases after epoch 50. The trend of $R_{true}$ reflects that the model is gradually selecting more and more support nodes that have exactly the same label as the query node. For example, the initial true label ratio for Cora's 2-way 5-shot problem is around 0.41 and it steadily increases to 0.8, indicating that only around 2 selected support samples in this task have false labels. This measure verifies that the proposed method can construct semantically correct meta-tasks even without label information.

5.4.2 *Analysis of the number of negatives.* Contrastive learning methods benefit from both the data augmentation and the large number of negative samples. Although COLA's loss has a similar form to supervised contrastive loss, the number of negative samples is relatively small. This is because all the negative samples of a node only come from the support set of other ways, e.g. $(N-1)k$ for a $N$-way $k$-shot problem. Consequently, we examine whether the meta-tasks constructed by COLA will benefit from a large number of negative samples just like contrastive learning does. Thus, we vary the number of negatives from $(N-1)k$ to $|\mathcal{V}|$ (number of nodes in the graph) and present the result in Figure 4c. We get a conclusion that is contrary to expectations: the performance of our model is negatively impacted by increasing the number of negative samples in each case. We conjecture that the advantages contrastive learning gains from a high number of negative samples do not transfer well to few-shot tasks. Consequently, it underscores the need for a unified method (COLA) that is more suitable for FSNC.

5.4.3 *Using all nodes and data augmentation indeed contributes to the success of COLA..* We evaluate whether the utilization of all nodes and data augmentation will be helpful in our model and

**Table 5: Results of w/ and w/o augmentations and nodes from $C_{test}$ on CiteSeer dataset.**

|  | $C \backslash C_{test}$ | $C_{test}$ | All nodes |
|---|---|---|---|
| w/ aug | 68.43 | 72.19 | 86.97 |
| w/o aug | 65.18 | 61.02 | 74.51 |

show the results of the CiteSeer dataset in Table 5. From the results, we can conclude that training without all nodes will lead to a performance decrease, especially when nodes from $C_{test}$ are not involved. We would like to emphasize that COLA's success is not solely due to the inclusion of test nodes. Despite other graph CL methods also using test nodes, COLA consistently surpasses them. This highlights COLA's unique strengths beyond just test node inclusion. Data augmentation is also important for our method since the meta-task construction relies on invariant graph information across the three augmented views. These findings underscore the fact that COLA significantly benefits from data augmentation, enabling the construction of meta-tasks that optimally leverage graph information.

## 6 CONCLUSION

In this paper, we focus on the transductive few-shot node classification. We first identify several key components behind the success of contrastive learning on FSNC, including the comprehensive use of graph nodes and the power of graph augmentations. We then introduce a new paradigm—**Co**ntrastive Few-Shot Node Cl**a**ssification (**COLA**). Unlike traditional meta learning methods that require label information, COLA finds semantically similar node embeddings to construct meta-tasks by leveraging the invariant information across three augmented graphs. COLA contains the advantages of both contrastive learning and meta learning on the few-shot node classification tasks. Through extensive experiments, we validate the essentiality of each component in our design and demonstrate that COLA achieves new state-of-the-art on all tasks. One limitation of our method is the increased computational cost due to the sorting operation used to find the support set, though this increase is linear to $|\mathcal{V}|$ and has no significant negative impacts in practice, which is detailed discussed in Appendix E. We believe our research will bring some new insights to the FSNC field.

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

# A    DATASET DESCRIPTION

We provide the dataset description in this section. The statistic descriptions are provided in Table 6. Note that two classes in Cora-Full dataset have extreme few samples, and cannot provide enough query set samples for validation or test, thus these two classes are omitted in all experiments.

**Cora** [37] is a citation network comprised of scientific publications that are categorized into one of seven classes. Here, each node represents a publication, with an edge indicating a citation relationship between two nodes. Each node features a binary word vector signifying the absence or presence of a word from a predefined dictionary.

**CiteSeer** [37] is also a citation network similar to Cora, where nodes represent scientific documents and edges represent citations.

**Amazon-Computer** [26] is a product co-purchasing network where nodes represent products and edges represent that the two products are frequently bought together. The label of each node is the product category it belongs to.

**Coauthor-CS** [26] is a collaboration network, where nodes represent authors and edges indicate the collaboration relationship between two authors. The node feature includes information about publishing history, affiliations, and research interests. The label represents the field of the author's research.

**CoraFull** [1] is also a citation network, which is extracted from the original data from the entire Cora network. Cora is a small subset of CoraFull.

**Ogbn-arxiv** [12] is a part of Open Graph Benchmark collection. It consists of the co-authorship derived from ArXiv, where nodes represent academic preprint papers and edges represent whether the two papers share at least one same author.

**Roman-Empire** [24] is a heterphilic graph dataset. This dataset is constructed by the Roman Empire article from English Wikipedia, where where each node represents one word and each edge represents whether the two nodes have a dependency.

We also present a histogram depicting the frequency of nodes within each category in Fig 5. As the figure illustrates, each dataset displays an imbalanced label distribution, with most exhibiting a long-tail distribution. This suggests that the complexity of each $N$-way $k$-shot meta-task will vary based on the sampled ways.

# B    REPRODUCIBILITY

We provide the **source code** of our method in this Github link: https://anonymous.4open.science/r/COLA-E4A0/README.md.

We report the average performance over 20 runs for each method and each task. All models were tested on a single NVIDIA A100 80GB GPU. The experiments on all baselines follow the setting of previous work TLP [30]. In our method COLA, we use GCN with ReLU activation function as the graph encoder for six homophilic dataset and use GraphSAGE as encoder for the heterphilic dataset: Roman-Empire. For all three augmentation operations, we take the combination of randomly dropping edge and dropping feature and the dropping ratio for Lookup and Support Embeddings are set to the same value. We implement Grid Search in [0.1, 0.2, 0.3, 0.4, 0.5, 0.6, 0.7] to obtain the augmentation ratio for each dataset. During the meta-train phase, we take the average loss function over 20 pseudo-meta-tasks to obtain the final loss function. During

the meta-test phase, we use the average performance of 100 meta-tasks and each meta-task includes 20 query samples for each way. Hyperparameters like learning rate, weight decay, and temperature parameter $\tau$ are presented in the YAML file of our provided codes.

# C    ADDITIONAL EXPERIMENTS

## C.1    Extra experiments of Section 3.1: ablation Study on GCL method with respect to nodes sampling and data augmentation

In Section 4.1, we delved into the influence of classes from which nodes are sampled, and the use of data augmentation within the Graph Contrastive Learning (GCL) method GRACE [41]. Here, we expand upon those results, providing further outcomes from additional datasets and tasks, with the aim of clarifying our preliminary findings. As the GRACE method employs all nodes in each optimization process, the experiment exceeds memory constraints with the ogbn-arxiv dataset; we present results for other datasets in the Figure 6.

Our conclusions are consistent across varied settings:

- Leveraging all nodes within the graph to compute contrastive loss leads to superior performance when compared to using only a subset of nodes.
- Interestingly, there are instances where using nodes exclusively from $C_{test}$ results in comparable, or even superior, performance relative to the scenario where nodes from $C_{train} \cup C_{val}$ are used. Despite the latter containing significantly more nodes than the former, incorporating nodes from test sets proves crucial in avoiding overfitting.
- In the majority of cases, the application of data augmentation techniques enhances model performance, given that the core of contrastive learning is to ascertain invariant information across distinct views. However, in a few scenarios, the integration of data augmentation has an adverse effect on model performance. We conjecture one reason might be that when the model inherently lacks adequate classification capability, introducing data augmentation equates to adding noise, thereby harming the effective learning of representation.

## C.2    Main results on Amazon-Computer, Coauthor-CS and ogbn-arxiv datasets

We present our main results on Amazon-Computer, Coauthor-CS and ogbn-arxiv datasets in Table 7. The results show that our method COLA still achieves the best performance in each task. The improvement is limited in some cases, e.g. 2-way 5-shot task of Amazon-Computer and the 5-way 5-shot task of Coauthor-CS, since the absolute accuracy is already close to 100%. In ogbn-arxiv dataset, we can notice that almost all the contrastive-based GCL methods face the out-of-memory (OOM) issue, but COLA will not since we only involve a small subset of nodes to construct meta-tasks.

# D    ABLATION STUDY ILLUSTRATION

We investigated three scenarios in Section 5.3.1 to verify the function of each component design. In this section, we first provide the

**Table 6: Dataset Description and Meta-task Class Split Ratio.**

|  | # of nodes | # of edges | # of feature | $C$ | $C_{train}$ | $C_{val}$ | $C_{test}$ |
|---|---|---|---|---|---|---|---|
| Cora [37] | 2,708 | 10,556 | 1,433 | 7 | 3 | 2 | 2 |
| CiteSeer [37] | 3,327 | 9,104 | 3,703 | 6 | 2 | 2 | 2 |
| Amazon-Computer [26] | 13,752 | 491,722 | 767 | 10 | 4 | 3 | 3 |
| CoraFull [1] | 19,793 | 126,842 | 8,710 | 70 | 38 | 15 | 15 |
| Coauthor-CS [26] | 18,333 | 163,788 | 6,805 | 15 | 5 | 5 | 5 |
| ogbn-arxiv [12] | 169,343 | 1,166,243 | 128 | 40 | 20 | 10 | 10 |
| Roman-Empire [24] | 22,662 | 32,927 | 300 | 18 | 8 | 5 | 5 |

**Table 7: Results on Amazon-Computer, Coauthor-CS and ogbn-arxiv datasets. (Top rows) Meta Learning. (Middle rows) Graph Contrastive Learning with fine-tuning. (Bottom row) COLA (our method). All scores are averaged over 20 runs. Evaluation metrics were scaled to 100 for readability purposes. In bold are methods with the best results for each task. In blue are methods with the best results in each group. *OOM* indicates out of memory.**

| Dataset | Amazon-Computer | | Coauthor-CS | | ogbn-arxiv | |
|---|---|---|---|---|---|---|
| Task | 2-way 1-shot | 2-way 5-shot | 5-way 1-shot | 5-way 5-shot | 5-way 1-shot | 5-way 5-shot |
| *Meta learning* | | | | | | |
| MAML [6] | 52.69 ± 2.23 | 59.19 ± 2.42 | 29.73 ± 1.54 | 43.78 ± 1.51 | 27.11 ± 1.49 | 28.83 ± 1.51 |
| ProtoNet [27] | 56.27 ± 2.54 | 63.11 ± 2.60 | 37.98 ± 1.69 | 51.10 ± 1.49 | 34.49 ± 1.72 | 46.21 ± 1.73 |
| Meta-GNN [40] | 60.54 ± 2.79 | 68.36 ± 3.15 | 54.17 ± 2.02 | 67.24 ± 1.56 | 27.42 ± 1.96 | 32.08 ± 1.65 |
| GPN [5] | 57.59 ± 2.67 | 74.86 ± 2.27 | 64.95 ± 1.43 | 75.42 ± 1.56 | 36.23 ± 1.48 | 48.85 ± 1.60 |
| G-Meta [13] | 62.56 ± 3.11 | 71.47 ± 2.97 | 59.87 ± 2.35 | 73.16 ± 1.40 | 26.45 ± 1.62 | 33.09 ± 1.65 |
| TENT [35] | 77.74 ± 3.16 | 86.06 ± 2.16 | 59.61 ± 1.87 | 74.84 ± 1.23 | 47.55 ± 1.93 | 61.98 ± 1.62 |
| *Graph Contrastive Learning ± Finetune* | | | | | | |
| BGRL [31] | 69.95 ± 3.15 | 83.99 ± 2.14 | 63.96 ± 2.19 | 89.53 ± 0.83 | 36.42 ± 1.70 | 53.63 ± 1.66 |
| MVGRL [11] | 65.95 ± 2.76 | 85.22 ± 2.08 | 69.64 ± 2.15 | 89.27 ± 1.04 | *OOM* | *OOM* |
| MERIT [15] | 77.35 ± 1.87 | 95.19 ± 0.69 | 85.74 ± 1.61 | 96.40 ± 0.39 | *OOM* | *OOM* |
| GraphCL [39] | 78.46 ± 3.05 | 93.53 ± 1.56 | 73.68 ± 2.49 | 89.74 ± 1.76 | *OOM* | *OOM* |
| GRACE [41] | 75.83 ± 2.84 | 88.46 ± 2.12 | 81.50 ± 1.88 | 92.24 ± 0.73 | *OOM* | *OOM* |
| SUGRL [21] | 85.49 ± 2.07 | 95.13 ± 0.89 | 92.47 ± 1.04 | 96.78 ± 0.33 | 57.46 ± 2.03 | 76.03 ± 1.38 |
| COLA (ours) | **87.52 ± 1.78** | **95.89 ± 1.02** | **93.23 ± 1.27** | **96.79 ± 0.68** | **60.41 ± 2.35** | **77.40 ± 2.09** |

illustration of these three scenarios for better understanding and then present the ablation study on other datasets.

- The first scenario is shown in Figure 7a, where we only have Query Embedding, thus both query and support sets are generated from Query Embedding itself.
- The second scenario is shown in Figure 7b. This scenario omits the Lookup Embedding, and the query embedding of the query node $v_i$ has to match with all node embeddings from Support Embedding.
- The third scenario is shown in Figure 7c. We discard the Support Embedding here, and the lookup embedding of the query node $v_i$ will match with all node embeddings from

Query Embedding. Thus, both query and support sets are from Query Embedding.

We then provide the component analysis results on Amazon-Computer, CoraFull, Coauthor-CS and ogbn-arxiv datasets in Table 8 and Table 9.

## E    LIMITATION

One limitation of COLA is the computational cost due to the way of meta-task construction, involving cosine similarity computation between the query node and all graph nodes, followed by a sort to obtain the top-$k$ nodes. Assuming a graph with $|\mathcal{V}|$ nodes and $|\mathcal{E}|$ edges, and node embeddings with dimension $d$. Given $t$ $n$-way

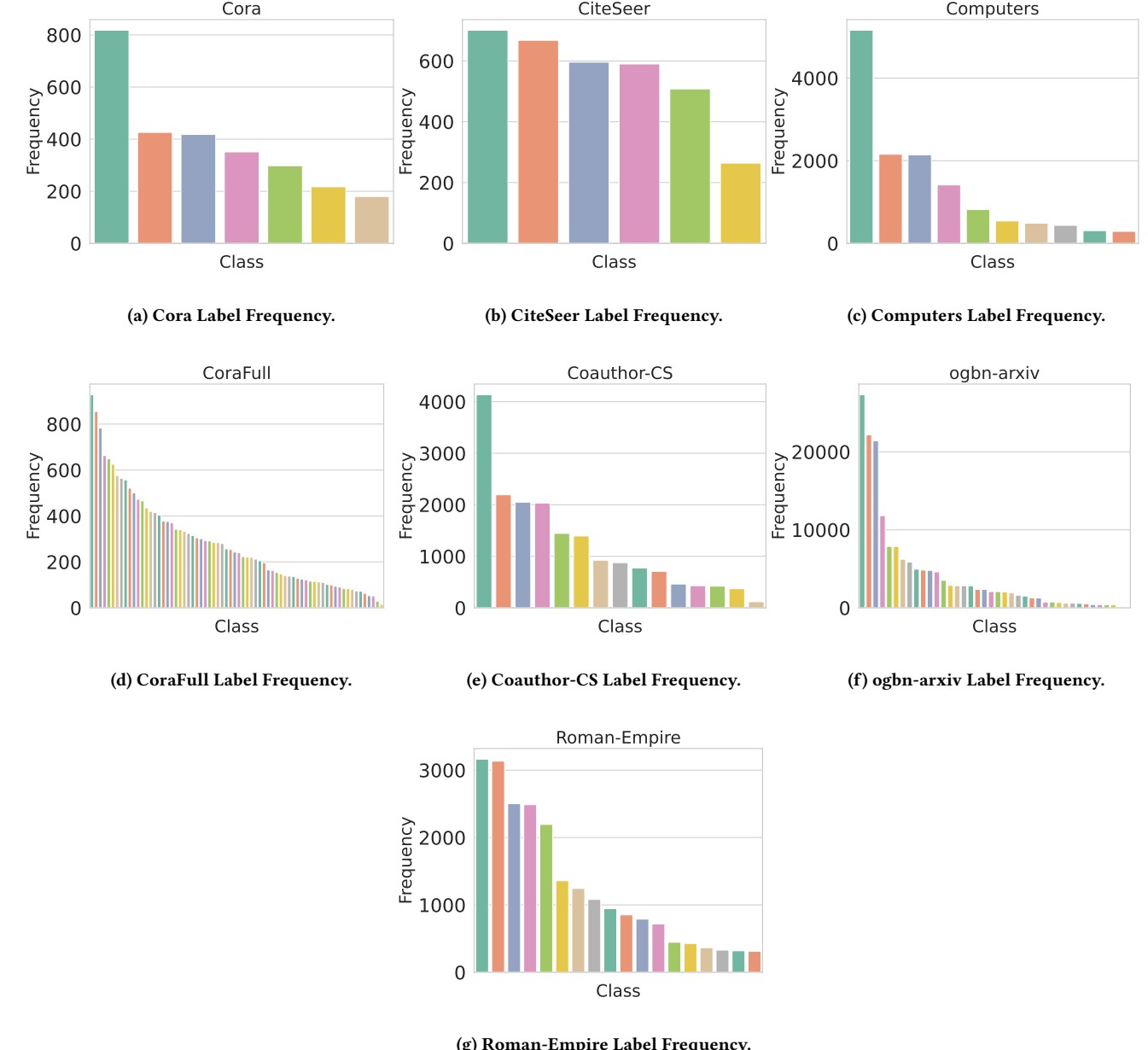

(a) Cora Label Frequency.

(b) CiteSeer Label Frequency.

(c) Computers Label Frequency.

(d) CoraFull Label Frequency.

(e) Coauthor-CS Label Frequency.

(f) ogbn-arxiv Label Frequency.

(g) Roman-Empire Label Frequency.

Figure 5: Label frequency distribution visualization of all datasets.

$k$-shot meta-tasks per training batch, the cosine similarity's time complexity is $O(|\mathcal{V}|dtn)$, that of sorting operation is $O(|\mathcal{V}|\sqrt{k}t)$, that of MPGNN is $O(|\mathcal{E}|)$. Thus, the time complexity of our method is $O(|\mathcal{V}|dtn + |\mathcal{E}|)$. Excluding the GNN, with $d, t, n$ all being constants, the complexity remains linear with respect to the number of nodes. We illustrate the convergence time (in seconds) across different datasets in Table 10. Although our convergence time is relatively longer than most baselines, this marginal increase is justifiable given the notable performance improvement.

Another limitation is that we do not explore much on different choices of the loss function and take the supervised contrastive loss [16] since in this work we focus more on the method to construct meta-tasks without labels. Future work could explore more on this aspect based on the meta-task construction.

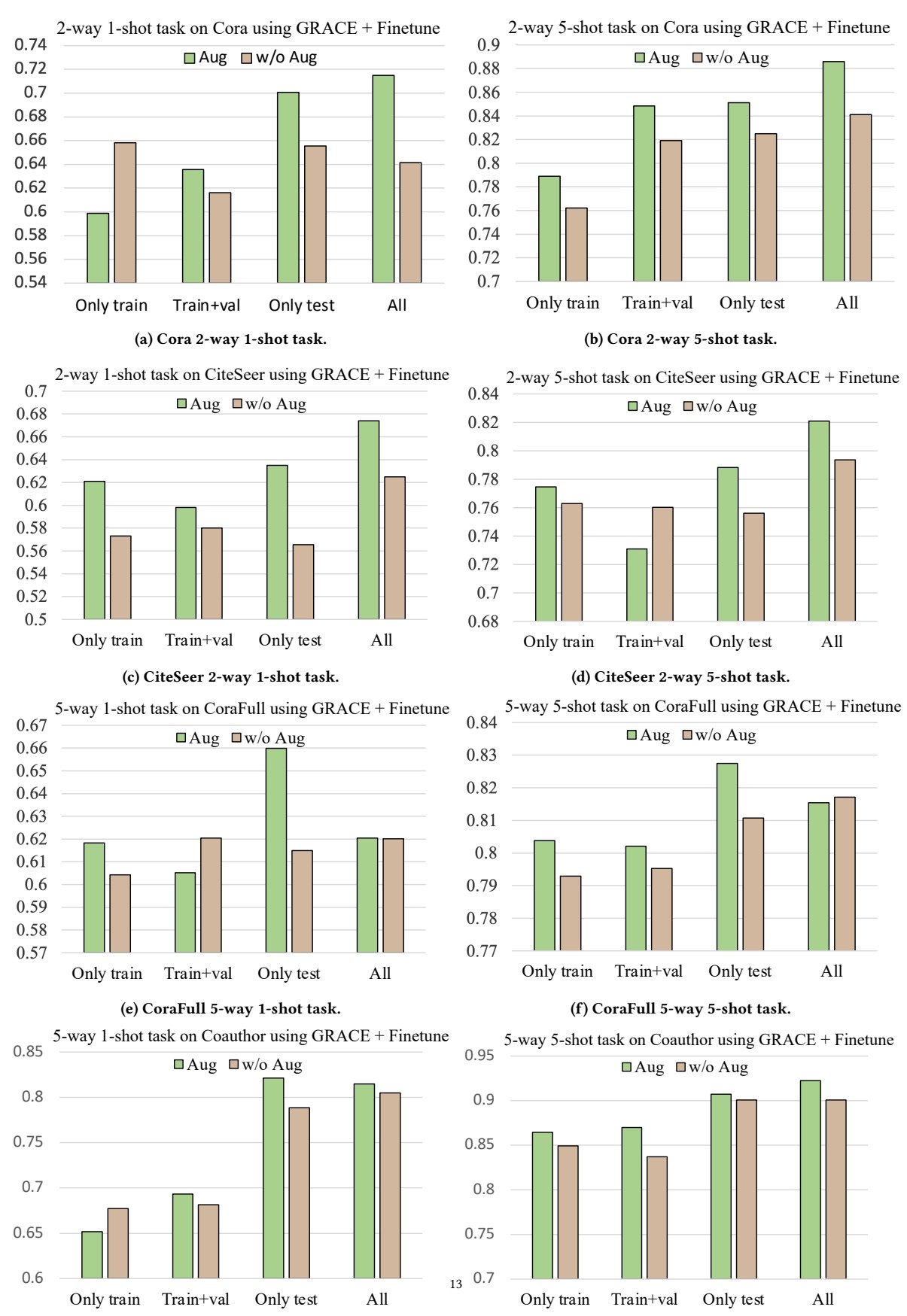

(a) Cora 2-way 1-shot task.

(b) Cora 2-way 5-shot task.

(c) CiteSeer 2-way 1-shot task.

(d) CiteSeer 2-way 5-shot task.

(e) CoraFull 5-way 1-shot task.

(f) CoraFull 5-way 5-shot task.

(g) Coauthor-CS 5-way 1-shot task.

(h) Coauthor-CS 5-way 5-shot task.

**Figure 6: Case study on GRACE+finetune framework.**

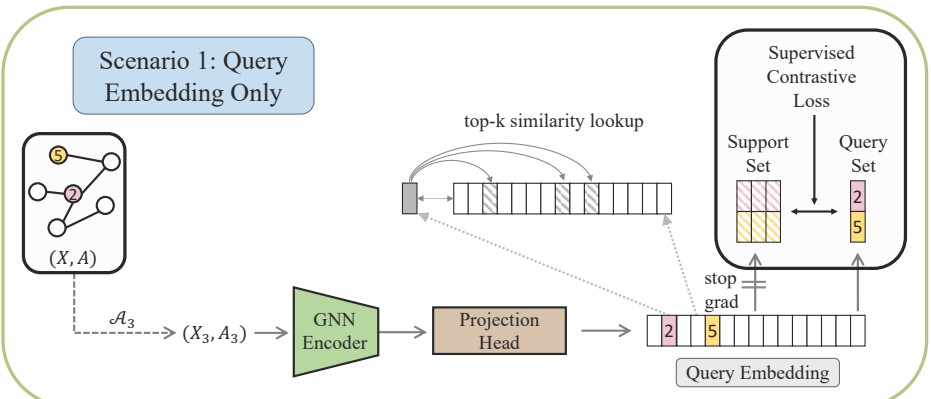

**(a) Scenario 1: Only Query Embedding.**

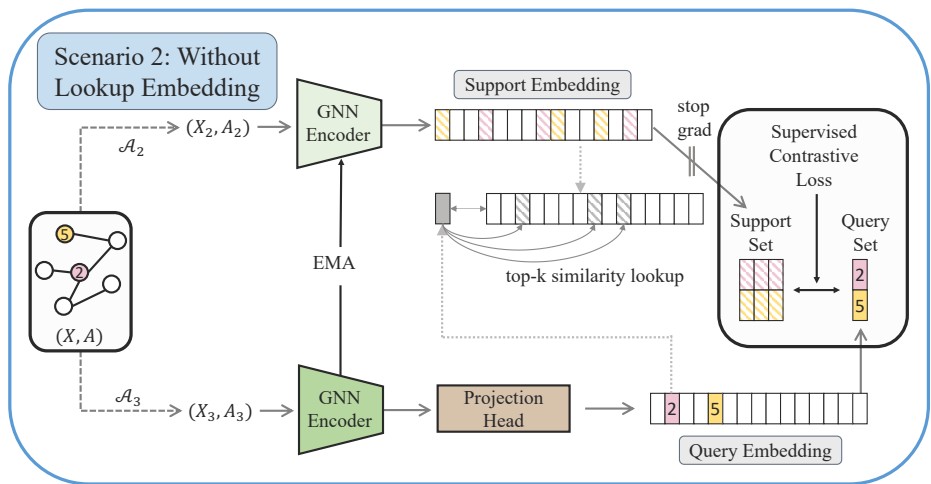

**(b) Scenario 2: Without Lookup Embedding.**

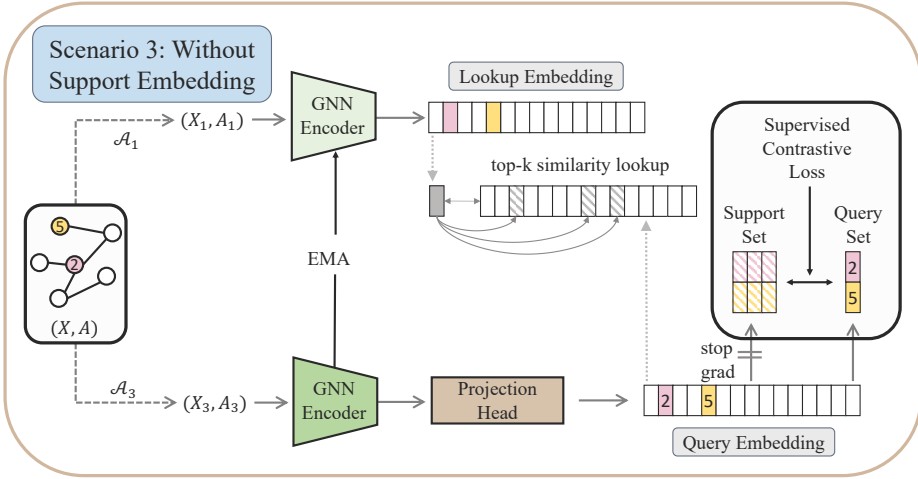

**(c) Scenario 3: Without Support Embedding.**

**Figure 7: Ablation Study Illustration.**

**Table 8: Component Analysis of Query (Q), Support (S), Lookup (L) Embeddings on Amazon-Computer and CoraFull datasets. The first three rows control different components in meta-task construction. The last row is COLA's setting. In bold are the best results, and underlines are the second best ones.**

| Q | S | L | Amazon-Computer | | | CoraFull | | |
|---|---|---|---|---|---|---|---|---|
| | | | 2-way 1-shot | 2-way 3-shot | 2-way 5-shot | 5-way 1-shot | 5-way 3-shot | 5-way 5-shot |
| ✓ | | | 71.04 ± 2.07 | 91.53 ± 2.26 | 92.76 ± 2.34 | 58.96 ± 1.99 | 76.48 ± 2.74 | 79.63 ± 2.39 |
| ✓ | ✓ | | 78.58 ± 2.61 | 85.87 ± 3.13 | 86.41 ± 2.45 | 64.62 ± 3.23 | 68.74 ± 2.10 | 71.43 ± 2.55 |
| ✓ | | ✓ | 80.06 ± 1.78 | 88.28 ± 2.33 | 90.37 ± 2.89 | 68.89 ± 2.09 | 75.17 ± 1.73 | 76.32 ± 2.76 |
| ✓ | ✓ | ✓ | **87.52 ± 1.78** | **93.08 ± 1.04** | **95.89 ± 1.02** | **74.36 ± 2.37** | **83.17 ± 2.48** | **86.59 ± 2.26** |

**Table 9: Component Analysis of Query (Q), Support (S), Lookup (L) Embeddings on Coauthor-CS and ogbn-arxiv datasets. The first three rows control different components in meta-task construction. The last row is COLA's setting. In bold are the best results, and underlines are the second best ones.**

| Q | S | L | Coauthor-CS | | | ogbn-arxiv | | |
|---|---|---|---|---|---|---|---|---|
| | | | 5-way 1-shot | 5-way 3-shot | 5-way 5-shot | 5-way 1-shot | 5-way 3-shot | 5-way 5-shot |
| ✓ | | | 80.37 ± 2.86 | 90.45 ± 1.38 | 93.57 ± 1.19 | 30.17 ± 2.36 | 54.57 ± 2.04 | 58.94 ± 3.01 |
| ✓ | ✓ | | 82.21 ± 3.43 | 84.90 ± 2.59 | 90.46 ± 1.76 | 42.49 ± 1.97 | 45.27 ± 2.00 | 49.68 ± 2.36 |
| ✓ | | ✓ | 88.75 ± 1.96 | 92.39 ± 1.73 | 94.53 ± 1.87 | 50.88 ± 2.73 | 53.96 ± 3.25 | 61.05 ± 2.84 |
| ✓ | ✓ | ✓ | **93.23 ± 2.17** | **96.42 ± 1.25** | **96.79 ± 0.68** | **60.41 ± 2.35** | **69.74 ± 2.28** | **77.40 ± 2.09** |

**Table 10: Convergence time comparison (in seconds) on a single NVIDIA A100 80GB GPU.**

| | Cora 2-way 1-shot | Cora 2-way 5-shot | CoraFull 5-way 1-shot | CoraFull 5-way 5-shot |
|---|---|---|---|---|
| MAML | 13.15 | 10.42 | 22.71 | 18.16 |
| ProtoNet | 17.40 | 16.83 | 31.39 | 19.38 |
| Meta-GNN | 26.33 | 25.03 | 92.99 | 83.32 |
| GPN | 13.3 | 10.67 | 34.43 | 53.04 |
| G-Meta | 46.62 | 191.82 | 196.01 | 662.54 |
| TENT | 64.46 | 43.90 | 58.12 | 58.92 |
| BGRL | 13.89 | 12.98 | 36.58 | 41.41 |
| MVGRL | 98.23 | 110.56 | 654.79 | 707.63 |
| MERIT | 955.60 | 1461.97 | 6240.12 | 8341.16 |
| GraphCL | 62.37 | 70.78 | 450.17 | 502.64 |
| GRACE | 8.38 | 6.80 | 74.42 | 41.53 |
| SUGRL | 25.05 | 16.07 | 542.86 | 428.57 |
| COLA | 83.43 | 103.64 | 619.65 | 817.43 |

