# OpenReview forum: "Graph Contrastive Learning Meets Graph Meta Learning: A Unified Method for Few-shot Node Tasks"
_ACM.org/TheWebConf/2024/Conference — TheWebConf24_

### Official Review · Reviewer_LSBT · 2023-11-16

**Novelty:** 5
**Technical Quality:** 5

**Review:**

Summary: This paper focuses on few-shot node classification, which has practical significance in real-world scenarios. Two important methods, i.e., meta-learning and contrastive learning on graphs are discussed. The strengths of contrastive learning over meta-learning consist of two aspects: comprehensive utilization of nodes, and more graph augmentation. They further propose a contrastive learning framework based on the N-way-K-shot paradigm in meta-learning.  The experiments on seven node classification datasets demonstrate the effectiveness of the proposed method.

Strengths:
1. he framework COLA is a simple but effective idea for few-shot graph learning.
2. Strong performance improvement over meta-learning baselines and other contrastive learning or graph data augmentation methods.
3. Detailed discussions and experimental comparisons with the methods for image datasets.

Weaknesses:
1. Although Table 2 has discussed different components and their effectiveness, my major concern is the rationality of using three augmentation views. It seems that the authors do not provide a theoretical analysis for this.
2. Some vague statements like “losing a significant portion of graph information” are in Line 111. What type of information does it lose?
3. Lack more discussion of graph few-shot learning in Related Work. Graph Few-shot Learning not only contains node-level tasks but also graph-level tasks. Some graph-level tasks can also be included, such as [1], [2]. Furthermore, I also encourage the authors to apply their method to graph-level tasks in experiments.

[1] FEW-SHOT LEARNING ON GRAPHS VIA SUPER-CLASSES BASED ON GRAPH SPECTRAL MEASURES, ICLR 20.

[2] Adaptive-Step Graph Meta-Learner for Few-Shot Graph Classification, CIKM 20.

**Questions:**

• Line150 said, “To achieve this, we use GNN, to get node embeddings from three augmented graphs. Is it necessary to use tree graphs? How much computation overload dese this operation need?
• Why is COLA better compared to the previous meta-learning method? Can you provide a deeper analysis for this? Line 331- 335 provides some explanations of the success of contrastive learning. My opinion is that each sample in contrastive learning can be viewed as a class, hence providing more enhanced task diversity. Am I right?
• What is the time complexity of this proposed transductive learning compared with meta-learning method like MAML?
• Why the improvement on heterophilous graph Roman-Empire is better than these homophilous graph datasets?
• Could you provide more discussion between the COLA and image contrastive learning methods, e.g., BFS, PsCo, from the method perspective?

**Reviewer Confidence:**

4: The reviewer is certain that the evaluation is correct and very familiar with the relevant literature

**Scope:**

4: The work is relevant to the Web and to the track, and is of broad interest to the community

---

### Official Review · Reviewer_ZPBk · 2023-11-17

**Novelty:** 5
**Technical Quality:** 4

**Review:**

The article introduces a new paradigm, Contrastive Few-Shot Node Classification (COLA), which combines graph contrastive learning with fine-tuning and meta-learning techniques. The empirical success of COLA, demonstrated through extensive experiments and achieving new state-of-the-art results on all tasks, underscores the significance of the proposed method. However, the article could benefit from providing more detailed insights into the experimental results, including comparisons with existing state-of-the-art methods. Additionally, addressing the limited understanding of why contrastive learning outperforms meta-learning, as mentioned in the introduction, would contribute to the overall impact of the work.
In summary, the proposed COLA framework introduces several innovative elements and demonstrates promising advantages, particularly in preventing overfitting and incorporating a broader range of graph information. However, further clarification and in-depth analysis of the experimental results would enhance the overall quality and impact of the article. Potential computational complexities and the need for further exploration of challenges and broader empirical validation should be considered.

Advantages:
1. Innovative Approach: The article introduces a novel paradigm, Contrastive Few-Shot Node Classification (COLA), which leverages the advantages of contrastive learning to enhance the existing meta-learning framework. This innovative approach addresses the question of whether contrastive learning can be effectively integrated into the meta-learning context for few-shot node classification.
2. Effective Use of Information: COLA effectively utilizes information from three augmented graphs to construct semantically correct meta-tasks without relying on label information. This approach ensures the comprehensive utilization of graph nodes and promotes semantic similarity-based construction of meta-tasks, which is different from conventional meta-tasks based on training labels.
3. Prevention of Overfitting: By constructing meta-tasks based on semantic similarity rather than training labels, COLA mitigates the risk of overfitting to training classes. This is a significant advantage, as it helps the model generalize better to new, unseen classes.
4. Incorporation of All Nodes: COLA meta-tasks involve all nodes in training, incorporating a more extensive range of graph information compared to traditional meta-learning methods that only use labeled nodes. This broad inclusion of nodes can enhance the model's understanding of the entire graph structure.

Disadvantages:
1. Complexity and Computational Cost: The proposed method involves the use of three augmented graphs and a GNN to construct meta-tasks, which may introduce computational complexity and increase the computational cost. This could be a potential drawback in scenarios where computational efficiency is crucial.
2. Limited Insight into Challenges: While the article outlines the process of constructing support sets in COLA, it could provide more insight into potential challenges or limitations associated with the proposed framework. A more thorough discussion of the challenges and how they are addressed would contribute to a better understanding of the method.
3. Empirical Validation Limited to Few Datasets: The article mentions extensive experiments on seven real-world datasets, but a more detailed discussion on the diversity and characteristics of these datasets would strengthen the empirical validation. Additionally, testing the proposed framework on a broader range of datasets could provide a more comprehensive assessment of its generalizability.

**Questions:**

Please see the details above.

**Ethics Review Description:**

None.

**Reviewer Confidence:**

4: The reviewer is certain that the evaluation is correct and very familiar with the relevant literature

**Scope:**

4: The work is relevant to the Web and to the track, and is of broad interest to the community

---

### Official Review · Reviewer_qkZ3 · 2023-11-23

**Novelty:** 6
**Technical Quality:** 5

**Review:**

The paper explores the important problem of the Few-Shot Node-Classification (FSNC) task through Meta-learning and Graph Contrastive Learning. Based on the recent works that discover the significant benefits of leveraging graph contrastive learning in few-shot node classification, the authors investigate further the reasons behind the advantages of graph contrastive learning. The authors propose to integrate the benefits of graph contrastive learning and meta-learning for few-shot node classification, named COLA, enabling the construction of meta-tasks without label information. The authors conduct extensive experiments to demonstrate the effectiveness of the proposed method.

**Questions:**

-Advantages.
(1) This paper is generally well-written and well-organized.
(2) The authors conduct extensive experiments to demonstrate the effectiveness of the proposed framework.
(3) The authors provide detailed analysis for the proposed method, regarding various components of their design. Furthermore, they analyze the reasons behind the benefits of graph contrastive learning.

-Disadvantages.
(1) The paper lacks sufficient comparisons of several more recent methods. For example, (1) “Contrastive Meta-Learning for Few-shot Node Classification”, KDD 2023. (2) “Graph few-shot learning with task-specific structures”, NeurIPS 2022. These methods are also based on the meta-learning framework and thus should be considered as baselines.

(2) The designed exepriments lack the consideration of scenaiors where only a small fraction of data is available. In graph-related tasks, a more general scenario would be only using limited labeled data.

(3) The graph contrastive learning methosd compared in this paper seem not to be state-of-the-art. That being said, there are more recent graph contrastive learning methods for comparisons.



Questions:

Please see Disadvantages.

**Reviewer Confidence:**

4: The reviewer is certain that the evaluation is correct and very familiar with the relevant literature

**Scope:**

4: The work is relevant to the Web and to the track, and is of broad interest to the community

---

### Official Review · Reviewer_Nwwf · 2023-11-30

**Novelty:** 3
**Technical Quality:** 5

**Review:**

This study proposes a few-shot node classification method by integrating contrastive learning and meta learning. The authors analyze why contrastive learning can be effective in this task, and concentrate on the main challenge to construct the support set in meta learning. Specifically, they achieve the final support set by incorporating three augmented graphs. The authors conduct extensive experiments to validate the performance of the proposed method. The results show that the method exceeds the SOTA baselines in most few-shot cases.

Overall, the paper is well-written and easy-to-follow. However, my major concern is that the novelty seems limited, since the proposed method can be treated as an application of contrastive learning for graph node classification.

**Questions:**

N/A

**Reviewer Confidence:**

3: The reviewer is confident but not certain that the evaluation is correct

**Scope:**

4: The work is relevant to the Web and to the track, and is of broad interest to the community

---

### Decision · Program_Chairs · 2024-01-22

**Decision:**

Accept

**Comment:**

The paper tackles a very important problem and the proposed method excels in novelty and performance. A number of adjustments and improvements in explanation and experiments have been made in responses to reviewers' suggestions.